# Distributional Concavity Regularization for GANs

**Shoichiro Yamaguchi, Masanori Koyama**
Preferred Networks
{guguchi, masomatics}@preferred.jp

## Abstract

We propose Distributional Concavity (DC) regularization for Generative Adversarial Networks (GANs), a functional gradient-based method that promotes the entropy of the generator distribution and works against mode collapse. Our DC regularization is an easy-to-implement method that can be used in combination with the current state of the art methods like Spectral Normalization and Wasserstein GAN with gradient penalty to further improve the performance. We will not only show that our DC regularization can achieve highly competitive results on ILSVRC2012 and CIFAR datasets in terms of Inception score and Fréchet inception distance, but also provide a mathematical guarantee that our method can always increase the entropy of the generator distribution. We will also show an intimate theoretical connection between our method and the theory of optimal transport.

## 1 Introduction

Generative Adversarial Networks (GANs) (Goodfellow et al., 2014) is a model consisting of two adversarial neural networks designed for the training of a generator distribution that mimics a target distribution defined on a (*often*) high dimensional space, and it has been successful in numerous applications including image and movie generations (Isola et al., 2017; Zhu et al., 2017; Saito et al., 2017). However, it has not yet completely established itself as a wildly scientific tool because of the sheer computational difficulty of its training process. As such, there has been numerous studies seeking for a way to stabilize the training process of GANs (Gulrajani et al., 2017; Miyato et al., 2018; Karras et al., 2018).

Mode collapse is a persistent central problem for the training of GANs, which collectively refers to the lack of diversity in generator distribution. For instance, without any countermeasure, GANs applied to multimodal mixture of Gaussians will often train a generator distribution with one mode (See Fig 22, Goodfellow (2016) for example). Mode collapse has been discussed in numerous literatures related GANs. To name a few, see Goodfellow (2016); Metz et al. (2016); Arjovsky & Bottou (2017); Arjovsky et al. (2017); Lin et al. (2017).

In words of statistical machine learning, mode collapse can be described as a case of entropy degeneration. A naive countermeasure against mode collapse is therefore to augment the entropy of the generator distribution. Not many studies to date, however, tackled the problem of mode collapse using this direct approach. Dai et al. (2017) partially realized this idea by actually evaluating the entropy of the generator distribution with variational inference and including it into the objective function. This type of strategy requires the user to continuously produce reliable estimates of the generators based on finite samples throughout the course of the training. As such, not much further improvement can be expected from this strategy, because precise estimation of the entropy tends to be computationally heavy, and empirical estimation is an excruciatingly difficult problem on its own in high dimension, as is the case in image and movie generation. In fact, the performance of classical methods based on kernel density estimation, for example, does not scale well with respect to dimension, and the number of samples required to control the MSE can grow exponentially with the dimension (Cacoullos, 1966; Ozakin & Gray, 2009).

In this study, we use the theory of *functional gradient* to develop a method that can promote the entropy of the generator distribution without directly estimating the entropy itself. All in all, the ob-

jective of GANs is find a *good* generator. The motive of functional gradient method is pay attention to the update of the generator in the *space of generator functions* itself, instead of the updates in the parameter space. In the function space, the functional properties of the generators are generally easier expressed than in the parameter space. As such, the search for a generator with *good* functional properties may be much easier in the space of functions, as opposed to the space of parameters. Throughout, this philosophy will serve as the basis of our method.

In more general technical term, the functional gradient of an objective function with respect to a model function is an *infinite dimensional gradient* computed over the infinite dimensional space of all models. In our case, our interest is the functional gradient of the GANs' objective function with respect to the generator function. As we will show, all variations of GANs to date are implicitly using the discriminator function to compute the *functional* derivative of the objective function with respect to the generator distribution function. In other words, the discriminator determines the direction to which the algorithm should move the mass of the current generator distribution; the discriminator determines what the next (target) distribution looks like. The work of Nitanda & Suzuki (2018) is a pioneer study that explicitly incorporated this idea into the training of neural generator models. Their study showed that one can carry out a faithful functional gradient-based update by inserting what they call *gradient layer* into the layers of neural generator function. Johnson & Zhang (2018) further polished this strategy by periodically distilling the networks.

A more precisely worded advantage of the functional gradient method is that, at every stage in the training process of the generator distribution, it allows the user to monitor what the next distribution (target distribution) looks like in terms of the current distribution. The ability of the functional gradient update to tell something about the next (target) distribution is a significant advantage of the functional gradient method over the conventional parametric methods, because the target distribution set forth by conventional parametric methods is usually expressed in a complicated parametric form (e.g. Deep Neural Nets (DNNs)), and its behaviors are often difficult to predict. This advantage of the functional gradient based-update suggests the possibility that we can control the update rule in order to deliberately direct the generator distribution to a distribution with preferred properties– which, in our case, include *high entropy*.

We discovered that, by locally concavifying the discriminator function, we can manipulate the functional gradient so that the next update for the generator will always target a distribution with higher entropy. From now on, we should refer to our method by *Distributional Concavity (DC) regularization*. Our method does not require the direct estimation of the entropy. We will not only show that our DC regularization can help improve the performance of GANs in terms of Inception score (Salimans et al., 2016) and Fréchet inception distance (FID) (Heusel et al., 2017), but also give a mathematical guarantee that our regularization will always increase the entropy of the generator distribution. We will also show that our method greatly outperforms the method of Dai et al. (2017) that directly estimates the entropy based on classically techniques.

The regularization strategy of monotonically increasing the entropy of the generator distribution over the course of its training is *not* without theoretical basis. Our DC regularization has close relations to the theory of optimal transport. We will show that, when the entropy of the true distribution is higher than the current generator distribution, the functional gradient that is properly derived from the optimal mass transport always increases the entropy of the distribution. Moreover, the functional update used in our method is always a monotonic mapping, which also turns out to be one of the properties satisfied by the update with optimal transport. This is a preferred property as well, because it tends to promote the smooth training of the generator. We summarize our contributions below:

- We propose an update for the generator of GANs that promotes the entropy of the generator without the need for an explicit estimation of the actual entropy.
- We show that, when the entropy of the true distribution is larger than that of the current generator distribution, the functional gradient derived from the 2-Wasserstein ($W_2$) optimal transport always increases the entropy of the distribution.
- We provide a mathematical guarantee that our method increases the entropy of the generator distribution at every step.
- We show that our method improves the results of GANs in terms of Inception score and FID score.

## 2 THEORY

### 2.1 GENERATIVE ADVERSARIAL NETWORKS

Let us first review the formulation of the original GANs. Unless otherwise noted, let us use boldface capital letters to refer to a random variable, and lower case letter for its realization. The training process of GANs (Goodfellow et al., 2014) is a two player min-max game in which the generator distribution $\mu_\theta$ is trained to minimize the *divergence* between the true distribution $\nu$ and the generator distribution $\mu_\theta$ measured by the current critic $F$, while the critic $F$ is trained to most strongly discriminate $\mu_\theta$ from $\nu$. Often, $\mu_\theta$ is produced by applying a paramatric generator function $G_\theta$ to a random seed variable with some distribution $\mu_z$, and it satisfies

$$E_{\mu_\theta}[h(\boldsymbol{X})] = E_{\mu_z}[h(G_\theta(\boldsymbol{Z}))] \tag{1}$$

for all measurable statistics $h$. In mathematical language, $\mu_\theta$ is a pushforward of $\mu_z$, and is often written as $G_\theta \# \mu_z$. In a nutshell, GANs aim to find an optimal parametric distribution $\mu_\theta$ that achieves the minimum value for

$$\min_{\mu_\theta} \max_F \mathbb{E}_\nu[\mathcal{L}_r(F(\boldsymbol{X}))] + \mathbb{E}_{\mu_\theta}[\mathcal{L}_g(F(\boldsymbol{X}))] := \min_{\mu_\theta} \max_F V(\mu_\theta, F) \tag{2}$$

where $\mathcal{L}_r$ and $\mathcal{L}_g$ are the functions of user's choice. We can retrieve the formulation of the original GANs (Goodfellow et al., 2014) by letting $\mathcal{L}_r(F(\boldsymbol{x})) = \text{softplus}(-F(\boldsymbol{x}))$ and $\mathcal{L}_g(F(\boldsymbol{x})) = \text{softplus}(F(\boldsymbol{x}))$, where $\text{softplus}(\cdot) = \log(1 + \exp(\cdot))$ There is a variety of choices for $(\mathcal{L}_r, \mathcal{L}_g)$ (Arjovsky et al., 2017; Lim & Ye, 2017; Mao et al., 2017; Nowozin et al., 2016). Rewritten as the optimization problem about the function $G$, our objective function is given by

$$\min_{\mu_\theta} \max_F \mathbb{E}_\nu[\mathcal{L}_r(F(\boldsymbol{X}))] + \mathbb{E}_{G\#\mu_z}[\mathcal{L}_g(F(\boldsymbol{X}))] := \min_{\mu_\theta} \max_F V(G, F) \tag{3}$$

### 2.2 FUNCTIONAL GRADIENT INTERPRETATION OF THE GANS UPDATE

Let us elaborate further on the mechanism of the functional-gradient based update and its connection to the conventional update of GANs . For ease of notation, let us write $L = \mathcal{L}_g \circ F$ in the equation (3), where the operator $\circ$ designates the function-composition. Let us remind ourselves that the objective of the min part of the min-max game in each step is to find a good update of $\mu_\theta$ that decreases $E_{\mu_\theta}[L(\boldsymbol{X})]$. It is no exaggeration to say that the choice of the next target distribution of $\mu_\theta$—or the distribution to which the $\mu_\theta$ will be updated—will completely determine the training efficiency of GANs. The most canonical and obvious approach is to use the information of the discriminator $L$ in making this decision. Note that, by appealing to a standard argument based on Taylor expansion, $L(\boldsymbol{x}) \geq L(\boldsymbol{x} - \alpha \nabla L(\boldsymbol{x}))$ for any value of $\boldsymbol{x}$ when $\alpha$ is suffiently small. Therefore, leaving the mathematical technicalities aside, one may expect that a carefully chosen $\alpha$ can ensure

$$E_{\mu_\theta}[L(\boldsymbol{X} - \alpha \nabla L(\boldsymbol{X}))] \leq E_{\mu_\theta}[L(\boldsymbol{X})]. \tag{4}$$

One naive suggestion based on this intuition is the update of $\mu_\theta$ to a distribution that can be constructed by taking a sample from $\mu_\theta$ (say, $\boldsymbol{x}$) and transporting it into the direction of $-\alpha \nabla L(\boldsymbol{x})$. Using the pushforward notation, this amounts to the update from $\mu_\theta$ to $(Id - \alpha \nabla) \# \mu_\theta$. The presumed relation (4) is equivalent to

$$E_{\mu_z}[L(G_\theta(\boldsymbol{X}) - \alpha \nabla L(G_\theta(\boldsymbol{X})))] \leq E_{\mu_z}[L(G_\theta(\boldsymbol{X}))],$$

and the corresponding suggested update in terms of the function $G_\theta$ is an update from $G_\theta$ to $(Id - \alpha \nabla L) \circ G_\theta$. That is, our suggestion amounts to the update from $G_\theta \# \mu_z$ to $((Id - \alpha \nabla L) \circ G_\theta) \# \mu_z$. Denoting

$$T_\alpha(\boldsymbol{x}) := \boldsymbol{x} - \alpha \nabla L(\boldsymbol{x}), \tag{5}$$

This is an update from $G_\theta \# \mu_z$ to $(T_\alpha \circ G_\theta) \# \mu_z$. This update of $G$ is called **functional gradient** update, and $T_\alpha$ is called a **transport function.** With enough (reasonable) regularity assumption of the function space and probability space, one can justify the argument we have made above using a differential calculus in an infinite dimensional space of functions(See appendix A). As a spoiler for what we will further elaborate later, our method regularizes $L$ so that the target distribution $(T_\alpha \circ G_\theta) \# \mu_z$ has higher entropy than $\mu_\theta = G_\theta \# \mu_z$.

We are, however, still not done in our description of a functional gradient perspective of the GANs update. In what we formulated above, there is no guarantee that the target distribution $(T_\alpha \circ G_\theta) \# \mu_z$

admits the same parametric representation as $G_\theta$; if $(T_\alpha \circ G_\theta)\#\mu_z$ cannot be expressed by the DNN that we have prepared for the training, all suggestions we have made above is for naught. The final remaining task in this update procedure is therefore to find the parameter $\theta^*$ such that $G_{\theta^*}\#\mu_z$ can best approximate the target distribution. Letting $\theta_{old}$ to denote the current choice of $\theta$ for the generator function, this can be done by solving the following optimization problem about $\theta$ :

$$\min_\theta \mathbb{E}_{\mu_z}\left[\|(T_\alpha \circ G_{\theta_{old}})(\boldsymbol{Z}) - G_\theta(\boldsymbol{Z})\|_2^2\right]. \tag{6}$$

The gradient of this sub-objective function (6) with respect to $\theta$ evaluated at $\theta = \theta_{old}$ is given by

$$2\mathbb{E}_{\mu_z}\left[\partial_\theta G_\theta(\boldsymbol{Z})\big((T_\alpha \circ G_{\theta_{old}})(\boldsymbol{Z}) - G_\theta(\boldsymbol{Z})\big)\right]\big|_{\theta=\theta_{old}} \tag{7}$$

$$= -2\alpha\mathbb{E}_{\mu_z}\left[\partial_\theta G_\theta(\boldsymbol{Z})|_{\theta=\theta_{old}}\nabla L(G_{\theta_{old}}(\boldsymbol{Z}))\right] \tag{8}$$

where $\partial_\theta$ designates the derivative operator with respect to $\theta$. This formulation is practically equivalent to the one introduced in xICFG (Johnson & Zhang, 2018). This turns out to be the familiar gradient update used in the usual GAN implementation. Indeed, in the usual implementation, the parameter for the generator is updated with the rule:

$$\theta_{new} = \theta_{old} - 2\alpha\nabla_\theta\mathbb{E}_{\mu_z}[L(G(\boldsymbol{Z};\theta))]\big|_{\theta=\theta_{old}} \tag{9}$$

$$= \theta_{old} - 2\alpha\,\mathbb{E}_{\mu_z}\left[\partial_\theta G_\theta(\boldsymbol{Z})|_{\theta=\theta_{old}}\nabla L(G(\boldsymbol{Z};\theta_{old}))\right] \tag{10}$$

In general, almost all variations of GANs to date (Arjovsky et al., 2017; Lim & Ye, 2017; Mao et al., 2017; Nowozin et al., 2016) uses this type of update rule for the training of the generator. In other words, all methods to date has been implicitly doing the functional-gradient type update all along, and the choice of $L$ has been determining the choice of the target distribution. As hinted a moment ago, this suggests that, by directly regularizing the $L$ in the usual update scheme of GANs, one can realize a functional gradient update of the generator with a controlled target distribution. This is the very gist of our algorithm.

### 2.3 CHOICE OF THE TARGET DISTRIBUTION

As inferred above, from the perspective of functional gradient, the min-max game of GANs can be decomposed into steps: (i) **the target construction step** that constructs the next *target distribution* using the functional gradient, and (ii) **the distillation step** that looks for a neural function that better approximates the target distribution. Now, the next natural pressing question is: "what type of $L$ should we use to make sure that the next target distribution is nice?"

As inferred in the introduction, the goal of this particular study is to create a sequence of updates that is less likely to suffer from *mode collapse*. Let us recall that, as long as we follow the standard update procedure of GANs, the choice of $L$ will entirely determine the property of the next target distribution. A proposal we would like to make in this study is to simply **choose the discriminator $L$ from the set functions that are concave on the support of the current distribution;**

> **Proposition 2.1.**
> *Let $\mu$ be a probability distribution on $\mathbb{R}^d$ .*
> *If $L$ is **concave** on the support of $\mu$, then $H(T_\alpha\#\mu) \geq H(\mu)$.*

Note that this statement is independent of the step size $\alpha$, because any positive scalar multiple of concave function is concave. This result ensures that any update with a concave $L$ will always increase the entropy; that is, the target distribution will be more *dispersed* than the current distribution, and the mode collapse is less likely to happen.

Additionally, our choice of $L$ guarantees that the transport function used to created the target distribution satisfies another *preferable* property called monotonicity. Monotonicity is a property that requires that there is no *crossings* in the transport:

$$\langle T(\boldsymbol{x}) - T(\boldsymbol{x}'), \boldsymbol{x} - \boldsymbol{x}'\rangle_{\mathbb{R}^d} \geq 0.$$

This is a preferred property because, as we will empirically show (see section 4.1) , the *crossings* in the transport tend to hinder the smooth training process. This is in fact somewhat intuitive, because crossings lead to wasteful transportation of the mass. In fact, this is a property that is achieved by the distributional update based on Optimal transport as well (Villani, 2008). Indeed, by the definition, the concavity of $L$ implies the strong convexity of $\frac{\boldsymbol{x}^2}{2} - \alpha L(\boldsymbol{x})$, which in turn implies

$$\langle T_\alpha(\boldsymbol{x}) - T_\alpha(\boldsymbol{x}'), \boldsymbol{x} - \boldsymbol{x}'\rangle_{\mathbb{R}^d} \geq \|\boldsymbol{x} - \boldsymbol{x}'\|_2^2 \geq 0, \tag{11}$$

which is a stronger condition than the monotonicity. Thus, by simply making $L$ concave, we can not only construct a target distribution whose entropy is greater than that of the current generator distribution, but also assure that the corresponding transport is monotonic. In a way, this is a statement about how complicated the function is, because the presence of the crossings imply the existence of a discontinuity or a region of *one to many* mappings. This is therefore a property that is likely to affect the distillation step. In fact, we will empirically show that the property of monotonicity affects the distillation step in positive way.

Indeed, we do not intend say that our suggestion for the property of $L$ is optimal—the user has the freedom to choose the set of properties to be required for $L$ so that the corresponding target distribution will have the desired properties that serves the purpose of the user.

## 2.4 RELATION TO OPTIMAL TRANSPORT

Our strategy for the distributional update that we have discussed so far has close relations to the optimal transport. In fact, an equation of the form (5) is ubiquitous in the theory of optimal transport. In this section, we will show the following:

1. By choosing $L$ to be concave, the transport $T_\alpha$ in the equation (5) becomes the optimal transport from the current distribution $\mu$ to the target distribution $T_\alpha\#\mu$.

2. If the entropy of the true distribution $\nu$ is greater than that of the current distribution $\mu$, any sequence of the updates of the distribution along the optimal transport from $\mu$ to $\nu$ increases the entropy monotonically. We can also ensure the monotonic increase in the entropy by using $T_\alpha$ with concave $L$.

3. By choosing $L$ to be concave, we can assure that the target distribution constructed from $T_\alpha$ will always be within $\alpha$ distance of the current distribution in Wasserstein ($W2$) sense.

In order to elaborate on these points, we would like to briefly introduce the notion of optimal transport. For a more rigorous introduction of the concept, please consult Villani (2008).
For $p \geq 1$, the $p$-Wasserstein distance $W_p(\mu, \nu)$ between two arbitrary distributions $\mu$ and $\nu$ with sufficient regularity is given by

$$\left(\inf_T E_\mu[\|\boldsymbol{X} - T(\boldsymbol{X})\|_2^p]\right)^{1/\mathrm{p}}, \tag{12}$$

where the inf above is taken over all measurable maps $T$ satisfying $T\#\mu = \nu$. When $p = 2$, it is known that the infimum of this cost function is achieved by $T^*$ that can be expressed as $\boldsymbol{x} - \nabla L^*$ for some $L^*$ that renders $\|\boldsymbol{x}\|^2/2 - L^*$ convex (Brenier, 1991). The search for $L^*$ and hence $T^*$ therefore amounts to the search for the closest coupling between $\mu$ and $\nu$ in the $L_2$ sense.

Another surprising fact is that the movement of the particle along the direction of $\nabla L^*$ monotonically decreases $W_2$ (Villani, 2008). That is, if $T_t^*(\boldsymbol{x}) = \boldsymbol{x} - t\nabla L^*(\boldsymbol{x})$, then $W_2(T_t^*\#\mu, \nu)$ monotonically decreases with $t$ (Villani, 2003). Please compare this transport equation with the equation (5). This $T_t^*$ is indeed the optimal transportation analogue of the functional gradient. The theorem in Brenier (1991) has a still surprising converse; any function $T$ that can be written as a gradient of a strictly convex function turns out to be *the* unique optimal transport from $\mu$ to $T\#\mu$. Using this fact, we can appeal to the proposition 2.2 below to claim that the functional update based on the optimal transport satisfies the following important property that shall be intuitively fulfilled if we are in fact moving the distribution toward the true distribution: if the current distribution has lower entropy than the true distribution, the sequence of the distributions produced by the *optimal transport based updates* should be monotonically increasing in the entropy:

---

**Proposition 2.2.**
*Suppose $\nu = T\#\mu$ for some $T$ that can be written as a gradient of a strictly convex function. If $H(\nu) \geq H(\mu)$, then $H(T_t\#\mu)$ is monotonically increasing on $t \in [0, 1]$.*

---

This proposition actually assures that the updates of DC regularization also satisfy the same property. Because we are designing the target distribution so that it will have higher entropy than the current distribution at every step, by letting $\nu$ in the proposition 2.2 to be the target distribution, we have a guarantee that the sequence of distributions constructed by our updates is also monotonically increasing in the entropy. To see this, simply note that we can automatically make $\|\boldsymbol{x}\|^2/2 - L$ to be convex by making $L$ concave,

The connection between our DC regularization and optimal transport is not limited to the property we introduced above. By the theory of Brenir, the choice $T_\alpha = Id - \alpha \nabla L$ with concave and 1-Lipschitz $L$ satisfies $W_2(\mu, T_\alpha \# \mu) = \sqrt{E[\|\alpha \nabla L(\boldsymbol{X})\|_2^2]} = \alpha$. Put in still other words, by constructing a target distribution with concave $L$, we can also assure that the target distribution is contained within the $W_2$ neighborhood of the current distribution function. This is a property that cannot be guaranteed with the conventional parameter-based updates. Monotonicity condition is also a property that is satisfied by the optimal transport. According the theory of Monge-Ampere (Villani (2008)), a transport map can be optimal only if it is monotonic. Thus, while not being exactly based on the optimal transport from the generator distribution to the true distribution, our $T$ shares many favorable properties in common with the optimal transport, and is very closely related to the theory of $W_2$ distance.

Reflecting on these facts, one might become tempted to say that we shall simply formulate GANs with the objective function that is solely based on $W_2$ distance between the generator distribution and the true distribution. However, as we will further articulate in the discussion section, the challenge remains to conduct faithful $W_2$-based updates in the implementation of GANs.

## 3 METHOD

If $\mu_\theta := G_\theta \# \mu_z$ is our current parametric generator distribution and $\nu$ is the true distribution, our DC regularization method first (i) proposes a target distribution $T \# \mu_\theta$ using a concave $L$, and then (ii) seek a measure $\mu_{\theta_{new}}$ that well approximates the target distribution $T \# \mu_\theta$. We use the following sampling-based penalty term in order to promote the concavity of $L$ ($= \mathcal{L}_g \circ F$):

$$\mathcal{L}_{dc}(F, \epsilon, \boldsymbol{x}_1, \boldsymbol{x}_2, d) = \max\{L(\epsilon\boldsymbol{x}_1 + (1-\epsilon)\boldsymbol{x}_2) - \epsilon L(\boldsymbol{x}_1) - (1-\epsilon)L(\boldsymbol{x}_2), d\}, \quad (13)$$

where $\boldsymbol{x}_1, \boldsymbol{x}_2$ are samples from the support of $\mu_\theta$, $\epsilon$ is a sample from the uniform distribution over $[0, 1]$, and $d$ is a positive scalar. Note that this term must be positive if $L$ is concave over the support of $\mu_\theta$. Our algorithm is summarized in Algorithm 1. The update rule in this algorithm uses the target distribution constructed by $T(\boldsymbol{x}) = \boldsymbol{x} - \nabla L$ with concave $L$. Intuitively speaking, the transport $T$ has an effect of moving $\mu_\theta$ toward $T \# \mu_\theta$ while dispersing the mass of $\mu_\theta$. See Fig 1 for a visual rendering of this interpretation.

In most practical application, the training of GANs begins by dispersing the initial distribution with small entropy and gradually molds the mass into what resembles the true distribution. Our transport ensures that this *dispersion* is consistently happening over the course of the training. As we will show in the result section, this regularization works in favor of the inception score without any noticeable downfall from *over-dispersion*, suggesting that the generator distribution created with the current state of the art techniques are still not *dispersed enough*.

---

**Algorithm 1** GANs algorithm with DC regularization

---

**for** each iteration **do**
    $\theta_{old} \leftarrow \theta$
    **the target construction step** : find $F^*$ that optimizes

$$\max_F V(G, F) + \lambda \mathbb{E}_{\text{Uniform}(\boldsymbol{\epsilon})} \mathbb{E}_{\mu_{\theta_{old}}(\boldsymbol{X}_1, \boldsymbol{X}_2)} [\mathcal{L}_{dc}(F, \boldsymbol{\epsilon}, \boldsymbol{X}_1, \boldsymbol{X}_2, d)] \quad (14)$$

    and construct $T_\alpha$ from $L := \mathcal{L}_g \circ F$ as in the equation (5).
    **the distillation step** : update generator by generator's objective

$$\min_\theta \mathbb{E}_{\mu_z} \left[ \|(T_\alpha \circ G_{\theta_{old}})(\boldsymbol{Z}) - G_\theta(\boldsymbol{Z})\|_2^2 \right] \quad (15)$$

    **end for**

---

## 4 EXPERIMENTAL RESULTS

We applied DC regularization to the training of GANs on CIFAR-10 (Torralba et al., 2008), CIFAR-100 (Torralba et al., 2008) and ILSVRC2012 dataset (ImageNet) (Russakovsky et al., 2015) in various settings and evaluated its performance in terms of Inception score (Salimans et al., 2016) and *Fréchet inception distance* (FID) (Heusel et al., 2017). Inception score and FID are performance measures that are commonly used to evaluate the severity of mode collapse. For the details of the evaluation, see Appendix C.1. We also conducted an additional set of experiments with artifical

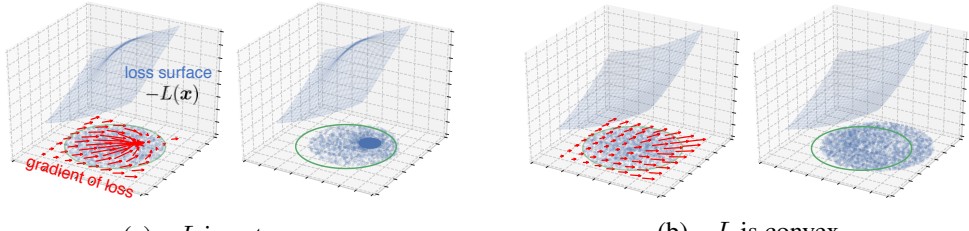

(a) $-L$ is not convex
(b) $-L$ is convex

Fig 1: The graph of $-L$ and its gradient vector field. Each point $x$ will be transported by $T$ along the direction of the vector field $-\nabla L$. Over the set on which $L$ is not concave, $T$ may move the points in the region come closer to each other. The use of concavified $L$ for the construction of transport function will make the points move away from each other.

dataset to investigate the properties of DC-regularization. We provide the results for additional experiments in the Appendix D as well.

### 4.1 INTRINSIC PROPERTIES OF DC REGULARIZATION

**Effect of DC regularization on entropy**
Using a simple Gaussian Mixture Model with five modes as the true distribution, we evaluated the sheer ability of the DC regularization to promote the entropy of the generator distribution. We used hinge loss for the objective function, and used DNNs to model both the discriminator and the generator. We trained the model with and without the DC regularization and reported the entropy of the Generator at different stages of the training by explicitly computing the determinant of the Jacobian at each layer. As one of the baseline, we trained EGAN-Ent-VI (Dai et al., 2017), which includes into its objective function a penalty against the negative entropy of the generator. The result is illustrated in Fig 2 (a). We see that the regularization is positively affecting the entropy at all stages of the training. Although to a less extent, we can confirm that our implementation of EGAN-Ent-VI is also preventing the degeneration of the entropy. Indeed, the persistent pressure to increase the entropy can result in over-dispsersed final product. However, this seems not to be a serious issue when it comes to the learning on big data like ImageNet. In terms of Inception score and FID score, all artificial generator distributions today are still far less diverse than the original dataset (Table 1), and there is still a large room left for the improvement of diversity.

**Effect of monotonicity in distillation step**
Recall that each round of GANs update consists of the target construction step and the distillation step. We conducted an experiment to verify the effect of the monotonicty on the distillation step. We will show a case in which, even if the target distribution constructed from a non-monotonic map is further away from the current distribution than the target distribution constructed from a monotonic map, the projection of the latter distribution onto the parametric function-space is much easier.
Let $K$ be a positive value. Starting from an initial distribution $X = G_{\theta_0}(Z)$, consider the following pair of maps to be applied to $X$:

$$T^{(m)}(x) = (x - K)\mathbb{1}_{(-1)^m x \leq 0} + (x + K)\mathbb{1}_{(-1)^m x \geq 0} \tag{16}$$

It is evident that $T^{(2)}$ is a monotonic mapping and $T^{(1)}$ is not. Denote the law of $X$ by $\mu$. We used $T^{(1)}\#\mu$ and $T^{(2)}\#\mu$ as target distributions, and trained the parameter $\theta$ using $O_m = E_Z[\|T^{(m)}(G_{\theta_0}(Z)) - G_\theta(Z)\|^2]$ as the objective function (Distillation). Note that this objective function takes the same value of $K^2$ for both $m = 1, 2$. Also, by the definition of the Wasserstein distance, $K = W_2(T^{(2)}\#\mu, \mu)$ while $W_2(T^{(1)}\#\mu, \mu)^2 = \inf_{T\#\mu=T_1\#\mu} E_\mu[\|T(X) - X\|^2] \leq E_\mu[\|T^{(1)}(X) - X\|^2] = K^2$. A naive intuition dictates that the distillation of $T^{(1)}\#\mu$ is easier, because $T^{(1)}\#\mu$ is distributionally closer to $\mu$ while the objective function evaluated at $T^{(1)}\#\mu$ is same as the evaluation at $T^{(2)}\#\mu$. However, as we can see in Fig 2 (b), the training about $O_2$ proceeds much faster than the training about $O_1$. This seemingly unintuitive observation can be supported by the theory of optimal transport. Note that $T_1$ is a derivative of $-K|X|$ and $T^{(2)}$ is a derivative of $K|X|$, and the latter is a convex function. This implies that $T^{(2)}$ is *the* optimal transport from $\mu$ to $T^{(2)}\#\mu$. This result (Fig 2 (b)) suggests that, when using the update rule similar to the equation 10, the training proceeds much faster when we choose a target distribution with monotonic mapping.

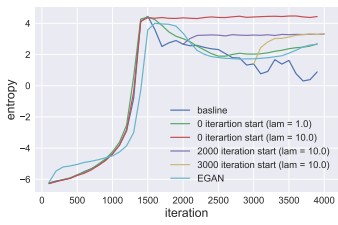

(a) Effect on entropy

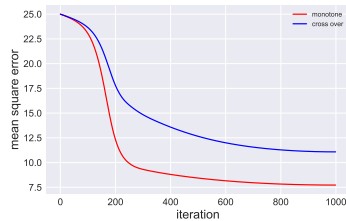

(b) Effect of monotonicity

Fig 2: The left graph plots the transition of the entropy of the generator distribution of GANs trained for the mixture of Gaussians. The marker "$k$ th iteration starts" designates the result for which our regularization was applied after $k$th step. We see that our method has the effect of preventing the entropy degeneration. The right graph plots the transitions of the energy for the distillation step when the target distributions were constructed by applying monotonic map (red) and non-monotonic map(blue) to a gaussian distribution. Notice that the distillation step converges faster when the target distribution is constructed from monotonic map.

## 4.2 RESULTS ON CIFAR-10 AND CIFAR-100

**Experiments with different architectures and objective functions**

We tested our algorithm for the training of GANs with six types of objective functions and two network architectures, and reported the performance on all 12 combinations. For the details of the objective functions and the architectures, please see Appendix C.2, C.3, C.4 For the training of the network, we applied Spectral Normalization (SN) (Miyato et al., 2018) to the full-connected layer and the convolution layer. Fig 3 and Fig 8 (in Appendix) respectively summarize Inception scores and FID for all 12 settings. We see that DC regularization is improving the performance irrespective of the choice of the architecture and the objective function.

**Experiments with different prior dimensions**

In general, without careful selection of the dimension of the prior distribution, the training of GANs tends to suffer a serious case of mode collapse. For image generation task, this will result in low inception score. We therefore applied DC regularization to the trainings of GANs on CIFAR-10 with very low prior dimensions as well as very large prior dimensions and evaluated the performance. For this experiment, we used GAN-variant2 (Appendix C.2) for the objective function and used SNDCGAN for the architecture. As for the experimental details, please see Appendix C.4. The results are summarized in Fig 4 and Fig 10 (in Appendix). When the prior dimension is below the inherent dimension of the dataset, the dimension of the generator distribution cannot match the dimension of the true distribution. Thus, the inception score of the generator trained with low dimensional prior is bound to be low, irrespective of the application of DC regularization. However, for $\dim(\boldsymbol{z}) \geq 5$, the inception score is as high as 7.5, and for all choices of $\dim(\boldsymbol{z})$, the generator trained with DC regularization consistently outperformed the generator trained without the DC regularization. Similar argument applies to the performance of DC regularization for high dimensional prior. Overall, the DC regularization provides some robustness against the choice of the prior dimension.

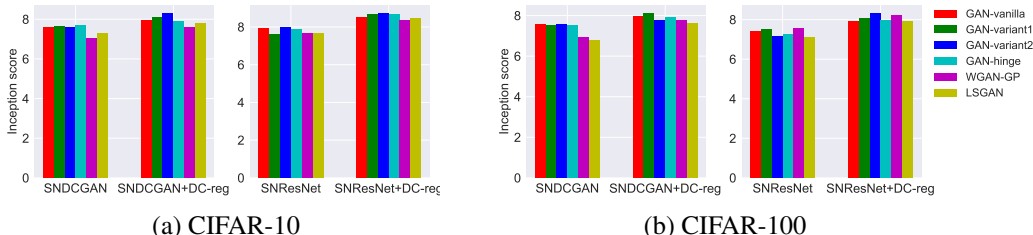

(a) CIFAR-10

(b) CIFAR-100

Fig 3: The inception scores of different GAN methods on CIFAR-10 and CIFAR-100 (higher the better). The right group of bars in each graph represents the set of scores achieved by the implementations with our regularization. The left group in each graph represents the set of scores achieved by the implementation without the regularization. Our regularization improves the inception score for all methods.

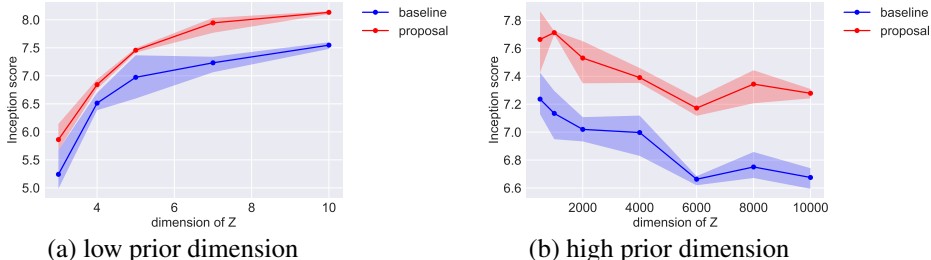

(a) low prior dimension          (b) high prior dimension

Fig 4: The performance of the DCGAN in terms of the inception score for CIFAR10 plotted against the dimension of $\mu_z$(standard Gaussian). The baseline is the DCGAN with spectral normalization(SN). We can see that too low a dimension and too high a dimension both negatively affect the performance. Notice that the DCGAN with DC regularization outperforms the baseline DCGAN for all extreme dimensions.

**Comparison with EGAN and other methods**

We compared the performance of our algorithm against EGAN-Ent-VI (Dai et al., 2017), another framework that can be used to control the entropy of the generator. We conducted this comparative study using SNDCGAN (Table 2) and SNResNet (Table 3), which uses Spectral Normalization that is known to have an effect of preventing the degeneration of the feature space. SNDCGAN is a method that is known to perform well on Cifar10 on its own. For the experimental setting of this study, please see the Appendix C.4. As we show in Table 1, DC regularization outperformed EGAN-Ent-VI on both models. We reported the result of EGAN-Ent-VI with spectral normalization, because it worked better than the version without SN. As we can see in the table 1, the EGAN with SN performed worse than the vanilla SNDCGAN. For high dimensional dataset like Cifar10, the variational inference for the negative entropy can be extremely difficult. It is possible that a poor variational inference in high dimensional space backfired for EGAN-VI's performance on Cifar10. We would also like to emphasize here that EGAN-Ent-VI requires a separate decoder in addition to the generator and the discriminator, and that our algorithm is easier to implement. For the full version of the Table and the visuals of the generated samples, please see Table 8 and Table 11 in the Appendix. We compared our algorithm against other competitive methods as well. The best performance of our method is almost on par with the state-of-the-art method (Karras et al., 2018). We are losing to progressive GAN (Karras et al., 2018) by a slight margin; we, however, would like to make a disclaimer that we are using much smaller architecture than Karras et al. (2018) for the performance evaluation of our method. We also confirmed that we can improve the result of Miyato et al. (2018) by using DC regularization together with SN. The results support that our method is helping the training process suppress *mode collapse* and is improving the overall performance. For the results of the experiments conducted with Wasserstein GAN with gradient penalty (WGAN-GP) (Gulrajani et al., 2017), please see the Fig 9 in Appendix.

### 4.3 RESULTS ON IMAGENET

Additionally, to evaluate our method's effectiveness on a higher dimensional dataset, we applied our method on ImageNet with 1000 classes, with each class containing approximately 1300 images. We compressed the images to $64 \times 64$ pixels prior to the experiments.

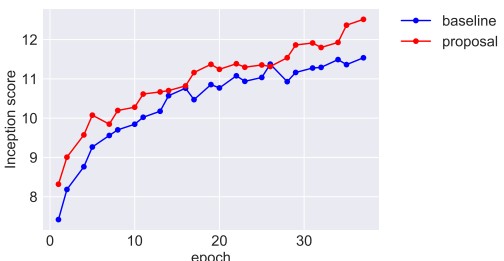

Fig 5: Performance of DC regularization on ImageNet. The baseline used in the comparison here is the GAN implemented with SN for Hinge-loss. The training with DC regularization achieves higher score consistently for all epochs.

For the objective function, we chose *GAN-hinge* (Appendix C.2), which was being used in Miyato et al. (2018). For the experimental settings, please see Appendix C.5 for the details. We can confirm on Fig 5 that our DC regularization is improving the Inception score throughout the course of the training.

Table 1: Inception scores and FIDs for unsupervised image generation on CIFAR-10 and CIFAR-100. The CIFAR-10 results for the models designated with † are cited from (Miyato et al., 2018), and the CIFAR-10 results with ‡ are cited from (Karras et al., 2018).

| Method | Inception score | | FID | |
|---|---|---|---|---|
| | CIFAR-10 | CIFAR-100 | CIFAR-10 | CIFAR-100 |
| Real data | 11.24 | 14.79 | 7.6 | 8.94 |
| EGAN-Ent-VI(SNDCGAN) | 6.95±.08 | 6.62±.10 | 29.0 | 33.3 |
| EGAN-Ent-VI(SNResNet) | 7.31±.12 | 6.67±.10 | 27.0 | 30.5 |
| **proposal** | | | | |
| SNDCGAN + DC reg | 8.08±.12 | 8.12±.11 | 24.6 | 25.8 |
| SNResNet + DC reg | 8.27±.08 | 8.27±.13 | 24.3 | 24.6 |
| SNResnetLarge + DC reg | 8.41±.10 | 8.20±.08 | 20.6 | 24.8 |
| SNResnetLarge-hinge + DC reg | 8.29±.09 | **8.41**±.11 | **19.5** | **23.6** |
| **baseline** | | | | |
| SNDCGAN-hinge† | 7.58±.12 | 7.57±.07 | 25.5 | 28.1 |
| SNResnetLarge-hinge† | 8.22±.05 | 7.54±.13 | 21.7 | 26.6 |
| Progressive GANs‡ | **8.56**±.06 | | | |

## 5 DISCUSSION

**Dilemma of GANs update**
As we have shown above, the experimental results support our claim that the DC regularization promotes the entropy of the generator distribution and stabilizes the training process of GANs. As mentioned briefly in the theory section 2, however, we do not have the guarantee that our update rule consistently reduces the $W_2$ distance between the generator distribution and the true distribution. This fault, however, is common to almost all GANs algorithms today. As we have shown, the conventional update rule is implicitly targetting a distribution of the form $T\#\mu$ where $T = Id - \alpha\nabla L$ for some $\alpha$ and a discriminator $L$. As it turns out, optimal transport takes this form only when we are optimizing $W_2$ distance, while the common constraint of $|\nabla L| = 1$ is a condition required for dual potential when $p = 1$ ($W_1$) (Villani, 2008). In other words, the conventional GANs are making the discriminator with $W_1$ criteria and updating the generator with $W_2$ criteria. If we are to faithfully create the discriminator with $W_2$ criteria, we must look for a Legendre-pair of dual potential functions. On the contrary, if we are to update the generator with $W_1$ criteria, one must look for a closed form solution for the $W_1$ transport. This, however, is in general a highly complex mathematical problem for which there is a separate field of study (Santambrogio, 2015). To the authors' best knowledge, no studies have provided a solid solution to this dilemma.

**Convexity vs Strong convexity**
We would like to also mention that our regularization is asking for more than what is required by $W_2$ theory. As mentioned above, in order for $T$ to be the optimal transport from $\mu$ to $T\#\mu$, $T$ only needs to be the gradient of a convex function. On the other hand, by asking $L$ to be concave, we are in fact asking for $T$ to be the gradient of a strongly convex function. This "overdo" is actually intentional. Recall that the parameter $\alpha$ in the transport $T_\alpha = Id - \alpha\nabla L$ corresponds to a step size in the update of the generator. If we train $L$ that only guarantees the convexity of $\|\boldsymbol{x}\|^2/2 - L(\boldsymbol{x})$, the functional update derived from such $L$ can be non-monotonic when the step size is large, because such $L$ only guarantees the convexity of $\|\boldsymbol{x}\|^2/2 - \alpha L(\boldsymbol{x})$ when $\alpha \leq 1$. Should we require $L$ to be concave, however, $\|\boldsymbol{x}\|^2/2 - \alpha L(\boldsymbol{x})$ is concave for any positive $\alpha$. In this context, we may therefore say that the DC regularization leaves much room to be playful about the learning schedule. Finally, as can be inferred from our proofs for the proposition 2.1 and 2.2, the strong convexity of $T$ has an effect of diffusing the mass of pdf in every direction. We can therefore expect the DC regularization to disperse the collapsed masses in the case of mode collapse. In the light of the fact that the training of GANs usually begins with dense distribution with small support, this dispersion effect should be helpful at the early stage of the training as well.

## ACKNOWLEDGMENTS

We would like to thank the members of PFN.Inc, particularly Daisuke Okanohara, Kouhei Hayashi, Masaki Watabnabe, Shin-ichi Maeda, Sosuke Kobayashi, Takeru Miyato, Kenta Oono, and Toshiki Kataoka for helpful comments and advices. We would also like to thank Atsushi Nitanda for constructive advices

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

# Appendix (Supplemental Material for Distributional Concavity Regularization for GANs)

## A   MORE MATHEMATICAL RENDITION OF SECTION 2

In this section, we re-explain the section 2 with more mathematical details. For ease of argument, we will assume the followings from here onward. First, as is the case in many application of GANs, we will assume that $\nu$ is a probability measure on the Euclidean space $\mathbb{R}^d$, and $\mu_z$ is a probability measure on some Euclidean space $U$. Let us also assume that $G_\theta : U \to \mathbb{R}^d$ is a parametric function of $\theta$ that is almost everywhere differentiable with respect to both input and $\theta$. We will first consider the optimization problem of the equation (3) with gradient descent about $\theta$. For ease of notation, let us write $L = \mathcal{L}_g \circ F$, where the operator $\circ$ designates the function-composition. We will assume that $L$ is a continuously differrentiable and uniformly bounded function on $\mathbb{R}^d$, and that the Euclidean norms of its gradient and operator norm of Hessian are also uniformly bounded. This type of regularity assumptions on the objective functions are used in other literatures as well (Nitanda & Suzuki (2018), Johnson & Zhang (2018)), and it is not too all unrealistic because the support of both target distributions and generator distribution are often compact throughout the course of the training in the real applications. We will also assume that the derivatives of all functions to appear are well defined and bounded on $\mathbb{R}^d$. The update rule of $\theta$ is given by:

$$\theta_{new} = \theta_{old} - \alpha\nabla_\theta\mathbb{E}_{\mu_z}[L(G(\mathbf{Z};\theta))]\big|_{\theta=\theta_{old}} \tag{17}$$

$$= \theta_{old} - 2\alpha\,\mathbb{E}_{\mu_z}\left[\partial_\theta G_\theta(\mathbf{Z})|_{\theta=\theta_{old}}\nabla L(G(\mathbf{Z};\theta_{old}))\right] \tag{18}$$

where $\partial_\theta$ designates the derivative operator with respect to $\theta$. In general, almost all variations of GANs to date (Arjovsky et al., 2017; Lim & Ye, 2017; Mao et al., 2017; Nowozin et al., 2016) uses this type of update rule for the training of the generator. Let us denote the law of $G_\theta(\mathbf{Z})$ (or $\mathbf{X}$) by $\mu_\theta$. Let $\mathcal{L}^2(\mu_\theta)$ denote the Hilbert space of $L^2$ integrable maps from $\mathbb{R}^d$ to itself, granted with the inner product $\langle a,b\rangle_{\mu_\theta} = E_{\mu_\theta}[a(\mathbf{X})^T b(\mathbf{X})]$. We will show that we can re-derive the equation (18) using the theory of functional gradient. To begin with, instead of updating the parameter $\theta$, we would like to consider directly updating the distribution $\mu_\theta$ by applying a $T \in \mathcal{L}^2(\mu_\theta)$ to a random variable generated from $\mu_\theta$. This will result in a new distribution $T\#\mu_\theta$, which is a unique distribution that satisfies $\mathbb{E}_{\mu_\theta}[h(T(\mathbf{X})] = \mathbb{E}_{T\#\mu_\theta}[h(\mathbf{X})]$ for all measurable $h$.

Now, in order to create an update rule for $T$, let us consider the Gâteaux derivative of $M(T) := E_{T\#\mu_\theta}[L(\mathbf{X})]$ with respect to $T$ into an arbitrary direction $\delta \in \mathcal{L}^2(\mu_\theta)$. In terse term, we would like to consider the effect of purturbing $T$ into the direction of $\delta$. By the compactnesss assumption on $\Omega$ together with the regularity assumptions for the functions defined hereof, we can appeal to the dominated convergence theorem and see that

$$\lim_{\epsilon\to 0}\frac{M(T+\epsilon\delta) - M(T)}{\epsilon} = \mathbb{E}_{\mu_\theta}[\nabla L(T(\mathbf{X}))^T\delta(\mathbf{X})]. \tag{19}$$

is uniform for all $\delta \in \mathcal{L}^2(\mu)$. This way, the term $\nabla L(T(\cdot))$ in the expression above is a Fréchet derivative, or the type of derivative to which the usual set of derivative rules can be applied. We will refer to this derivative as *functional derivative* for short. Note that the *directional functional derivative* $\mathbb{E}_{\mu_\theta}[(\nabla L(T(\mathbf{X})) \cdot \delta)(\cdot)]$ will always take a positive value when $\delta \propto \nabla L(T(\cdot))$. Thus, for $\alpha$ small enough, an update from the current choice of $T$ to $T - \alpha\nabla L \in \mathcal{L}^2(\mu_\theta)$ will decrease the objective function. Here, we are interested in the *derivative* computed at $T = Id$, so the transformation we would like to apply to $\mu_\theta$ is

$$T_\alpha(\mathbf{x}) = \mathbf{x} - \alpha\nabla L(\mathbf{x}). \tag{20}$$

This is indeed a transportation of a mass into the direction of $\alpha\nabla L$. Now, for the update of $\theta$, we would like to design $\theta_{new}$ such that $\mu_{\theta_{new}}$ is closer to the **target distribution** $T\#\mu_{\theta_{old}}$ than $\mu_{\theta_{old}}$. That is, we would like to choose $\theta$ that minimizes

$$\min_\theta \mathbb{E}_{\mu_z}\left[\|(T_\alpha \circ G_{\theta_{old}})(\mathbf{Z}) - G_\theta(\mathbf{Z})\|_2^2\right]. \tag{21}$$

The gradient of this sub-objective function with respect to $\theta$ is given by

$$2\mathbb{E}_{\mu_z}\left[\partial_\theta G_\theta(\mathbf{Z})\big((T_\alpha \circ G_{\theta_{old}})(\mathbf{Z}) - G_\theta(\mathbf{Z})\big)\right]\big|_{\theta=\theta_{old}} \tag{22}$$

$$= -2\alpha\mathbb{E}_{\mu_z}\left[\partial_\theta G_\theta(\mathbf{Z})|_{\theta=\theta_{old}}\nabla L(G_{\theta_{old}}(\mathbf{Z}))\right]. \tag{23}$$

Evaluating this at $\theta = \theta_{old}$, we recover the gradient used in the equation (18). This formulation is practically equivalent to the one introduced in xICFG (Johnson & Zhang, 2018).

## B  THE PROOFS FOR THE PROPOSITIONS 2.1 AND 2.2

We will first prove the proposition 2.2. Let us begin with a useful lemma. We will assume that all regularity assumptions made in Section 2 holds for all variables and functions that appear in this section. Also, we will use the $Df$ to denote the derivative of $f$. Also assume that, unless otherwise noted, $Df(x)$ indicates $D_x f(x)$, the derivative of $f$ with respect to $x$.

**Lemma B.1.** *Let $\mu$ and $\nu$ be probability distributions on $\mathbb{R}^n$, and suppose that we can write $(Id - \nabla L)\#\mu = \nu$ with one-to-one $\nabla L$. Let us also write $T = (Id - \nabla L)$, and let $T_t = Id - t\nabla L$ be its time-linear interpolation. Then, for all $t \in [0, 1]$,*

$$\frac{d}{dt}H(T_t\#\mu) = -E_\mu\left[tr(I - tD^2L(\boldsymbol{X}))^{-1}(D^2L(\boldsymbol{X}))\right]. \tag{24}$$

*Proof.* Let $p_\mu$ be the pdf of $\mu$. If $\boldsymbol{y} = T(\boldsymbol{x})$, by the one-to-one assumption $\det(DT(\boldsymbol{x})) > 0$ and we may let $d\boldsymbol{y} = \det(DT(\boldsymbol{x}))d\boldsymbol{x}$. By appealing to the fact that $E_\mu[h(T(\boldsymbol{X}))] = E_\nu[h(\boldsymbol{Y})]$ for arbitrary $h$,

$$\int h(T(\boldsymbol{x}))p_\mu(\boldsymbol{x})d\boldsymbol{x} = \int h(\boldsymbol{y})p_{T\#\mu}(\boldsymbol{y})dy = \int h(T(\boldsymbol{x}))p_{T\#\mu}(T(\boldsymbol{x}))\det(DT(\boldsymbol{x}))d\boldsymbol{x} \tag{25}$$

and we can deduce the identity $p_{T\#\mu}(T(\boldsymbol{x}))\det(DT(\boldsymbol{x})) = p_\mu(\boldsymbol{x})$. Building on this fact, with straightforward computation we can say

$$H(T_t\#\mu) = -E_{T_t\#\mu}[\log p_{T_t\#\mu}(\boldsymbol{X})] \tag{26}$$

$$= -\int p_\mu(\boldsymbol{x})\log p_{T\#\mu}(T_t(\boldsymbol{x}))d\boldsymbol{x} \tag{27}$$

$$= -\int p_\mu(\boldsymbol{x})\left(\log p_\mu(\boldsymbol{x}) - \log\det(DT_t(\boldsymbol{x}))\right)d\boldsymbol{x} \tag{28}$$

$$= H(\mu) + E_\mu\left[\log\det(DT_t(\boldsymbol{X}))\right]. \tag{29}$$

Assuming that the distributions are regular enough that we can swap the integral, we can appeal to Jacobi's formula and deduce

$$\frac{d}{dt}H(T_t\#\mu) = \frac{d}{dt}E_\mu\left[\log\det(DT_t(\boldsymbol{X}))\right] \tag{30}$$

$$= \frac{d}{dt}E_\mu\left[\log\det(I - tD^2L(\boldsymbol{X}))\right] \tag{31}$$

$$= E_\mu\left[\frac{\det(DT_t(\boldsymbol{X}))\mathrm{tr}(I - tD^2L(\boldsymbol{X}))^{-1}(-D^2L(\boldsymbol{X}))}{\det(DT_t(\boldsymbol{X}))}\right] \tag{32}$$

$$= -E_\mu\left[\mathrm{tr}(I - tD^2L(\boldsymbol{X}))^{-1}(D^2L(\boldsymbol{X}))\right]. \tag{33}$$

$\square$

Now, let us prove the proposition 2.2. Without loss of generality, choose $L$ so that $DT = \frac{\|\boldsymbol{x}\|^2}{2} - L$. Assuming that $D^2L$ is diagonalizable and that its spectrum is uniformly bounded away from 1, let $\mathrm{spec}(D^2L) = \{\lambda_k\}$ and write $\mathrm{diag}\{\lambda_k\}_k = \Lambda$. Because $\frac{\|\boldsymbol{x}\|^2}{2} - L$ is assumed to be strictly convex, $D^2\left(\frac{\|\boldsymbol{x}\|^2}{2} - L\right) = I - D^2L$ is positive definite, and $1 - \lambda_k(\boldsymbol{x}) > 0$ for all $k$. Appealing to the line equation (29) we see that the assumption $H(\nu) > H(\mu)$ is equivalent to

$$E_\mu\left[\log\det(T(\boldsymbol{X}))\right] = E_\mu\left[\log\det(I - D^2L(\boldsymbol{X}))\right]$$

$$= E_\mu\left[\sum_k \log(1 - \lambda_k(\boldsymbol{X}))\right]. \tag{34}$$

$$> 0$$

Let us write $max_k\{1 - t\lambda_k(\boldsymbol{x})\} = C_t(\boldsymbol{x}) > 0$. Using the assumption that the support of $\mu$ is compact, we can also say $0 < \max_{\mathrm{supp}(\mu)}\{C_t(\boldsymbol{x})\} := c_t < \infty$. In general, if $A$ is positive definite and diagonalizable, with straightforward argument we can say

$$\log(\det A) \leq \mathrm{tr}(A - I) \tag{35}$$

Using this fact, we see that

$$0 < E_\mu \left[ \log \det(I - D^2 L(\boldsymbol{X})) \right] \tag{36}$$

$$\leq E_\mu \left[ \mathrm{tr}(-D^2 L(\boldsymbol{X})) \right] \tag{37}$$

$$= E_\mu \left[ \sum_k -\lambda_k(\boldsymbol{X}) \right] \tag{38}$$

Suppose that $D^2 L$ is diagonalizable with the change of basis matrix $P$. Then

$$-E_\mu \log \left[ \mathrm{tr} \left( (I - tD^2 L(\boldsymbol{X}))^{-1} D^2 L(\boldsymbol{X}) \right) \right] = -E_\mu \left[ \mathrm{tr} \left( (P^{-1}(I - t\Lambda)P)^{-1} P^{-1}\Lambda(\boldsymbol{X})P) \right) \right] \tag{39}$$

$$= -E_\mu \left[ \mathrm{tr} \left( P^{-1}(I - t\Lambda)^{-1} PP^{-1}\Lambda(\boldsymbol{X})P \right) \right] \tag{40}$$

$$= -E_\mu \left[ \mathrm{tr} \left( (I - t\Lambda(\boldsymbol{X}))^{-1}\Lambda(\boldsymbol{X}) \right) \right] \tag{41}$$

$$= -E_\mu \left[ \sum_k \frac{\lambda_k(\boldsymbol{X})}{1 - t\lambda_k(\boldsymbol{X})} \right] \tag{42}$$

$$\geq -E_\mu \left[ \sum_k \frac{\lambda_k(\boldsymbol{X})}{c_t} \right] \tag{43}$$

$$> 0 \tag{44}$$

This concludes the proof of the proposition 2.2. The proposition 2.1 is much simpler to prove. In order to assure the $H(\nu) \geq H(\mu)$, we only need to guarantee that $E_\mu \left[ \log \det(DT(\boldsymbol{X})) \right] \geq 0$. By requiring $L$ to be concave, we would make $\|\boldsymbol{x}\|^2/2 - L(\boldsymbol{x})$ to be convex so that the eigenvalues of $DT$ will all become positive. The result then follows trivially from the argument similar to the one that leads to equation (34).

## C  EXPERIMENTAL SETTINGS

### C.1  PERFORMANCE MEASURES

For the measure of GANs' performance on the image dataset, we used Inception score (Salimans et al., 2016). Inception score was introduced originally as an exponentiated divergence measure based on the trained Inception convolutional neural network (Szegedy et al., 2015), which is often called *Inception Model*. Using $p(y|\boldsymbol{x})$ to denote the Inception model, the inception score is given by $I(\{\boldsymbol{x}_n\}_{n=1}^N) := \exp(\hat{\mathbb{E}}[D_{\mathrm{KL}}[p(y|\boldsymbol{x})||p(y)]])$, where $\hat{E}$ indicates the empirically approximated expectation. Often times, $p(y)$ is approximated with $\frac{1}{N}\sum_{n=1}^N p(y|\boldsymbol{x}_n)$.

The dominating consensus among the machine learning community is that this score is strongly correlated with subjective human judgment of image quality. Following the procedure in Salimans et al. (2016), we generated 5000 examples from each trained generator and calculated the Inception score on the samples. We evaluated the score 10 times with different seeds for the generation of $x_n$ and reported the average and the standard deviation of the scores.

Fréchet inception distance (FID) (Heusel et al., 2017) is another measure for the quality of the generated examples that uses 2nd order information of the final layer of the inception model. The FID is based on *Frechet distance* (Dowson & Landau, 1982) (Not to be confused with FID), which is the 2-Wasserstein distance between two distribution multivariate Gaussian distributions, $p_1$ and $p_2$. The Wasserstein distance for two Gaussian distributions have a closed form, and is given by

$$F(p_1, p_2) = \|\boldsymbol{\mu}_{p_1} - \boldsymbol{\mu}_{p_2}\|_2^2 + \mathrm{trace}\left( C_{p_1} + C_{p_2} - 2(C_{p_1} C_{p_2})^{1/2} \right), \tag{45}$$

where $\{\boldsymbol{\mu}_{p_1}, C_{p_1}\}$, $\{\boldsymbol{\mu}_{p_2}, C_{p_2}\}$ are the mean and covariance of samples from $q$ and $p$, respectively. Now, if $f_\ominus$ is the output of the final layer of the inception model before the softmax, the Fréchet inception distance (FID) between two distributions $p_1$ and $p_2$ on the images is the *Fréchet distance* distance between $f_\ominus \circ p_1$ and $f_\ominus \circ p_2$. We empirically computed the Fréchet inception distance between the true distribution and the generated distribution using 10000 samples from the true distribution and 5000 samples from the generator distribution.

## C.2 GAN's objective function

We used applied DC regularizaiton to the training with the following set of objective functions ($\text{softplus}(x) = \log(1 + \exp(x))$).

**GAN-vanilla**     (Goodfellow et al., 2014)

$$\min_G \max_F \mathbb{E}_\nu[\text{softplus}(F(\boldsymbol{X}))] + \mathbb{E}_{\mu_z}[\text{softplus}(-F(G(\boldsymbol{Z})))] \tag{46}$$

**GAN-variant1**     (Goodfellow et al., 2014)

$$\max_F \mathbb{E}_\nu[\text{softplus}(F(\boldsymbol{X})] + \mathbb{E}_{\mu_z}[\text{softplus}(-F(G(\boldsymbol{Z})))] \tag{47}$$

$$\min_G \mathbb{E}_{\mu_z}[-\text{softplus}(F(G(\boldsymbol{Z})))] \tag{48}$$

**GAN-variant2**

$$\max_F \mathbb{E}_\nu[\text{softplus}(F(\boldsymbol{X})] + \mathbb{E}_{\mu_z}[\text{softplus}(-F(G(\boldsymbol{Z})))] \tag{49}$$

$$\min_G \mathbb{E}_{\mu_z}[-F(G(\boldsymbol{Z}))] \tag{50}$$

**GAN-hinge**     (Lim & Ye, 2017)

$$\max_F \mathbb{E}_\nu[\min(0, -1 + F(\boldsymbol{X}))] + \mathbb{E}_{\mu_z}[\min(0, -1 - F(G(\boldsymbol{Z})))] \tag{51}$$

$$\min_G \mathbb{E}_{\mu_z}[-F(G(\boldsymbol{Z}))] \tag{52}$$

**WGAN-GP**     (Gulrajani et al., 2017)

$$\max_F \mathbb{E}_\nu[F(\boldsymbol{X})] - \mathbb{E}_{\mu_z}[F(G(\boldsymbol{Z}))] + \lambda \mathbb{E}_{\hat{\mu}}[(\|\nabla_{\hat{\boldsymbol{X}}} F(\hat{\boldsymbol{X}})\|_2 - 1)^2] \tag{53}$$

$\hat{\mu}$ is the law of $ux + (1-u)z$, with $(x, z, u) \sim \mu \times \nu \times U[0,1]$.

$$\min_G \mathbb{E}_{\mu_z}[-F(G(\boldsymbol{Z}))] \tag{54}$$

**LSGAN**     (Mao et al., 2017)

$$\min_F \mathbb{E}_\nu[((F(\boldsymbol{X}) - 1)^2] + \mathbb{E}_{\mu_z}[(F(G(\boldsymbol{Z})) + 1)^2] \tag{55}$$

$$\min_G \mathbb{E}_{\mu_z}[(F(G(\boldsymbol{Z})) - 1)^2] \tag{56}$$

**Feature Matching**     (Salimans et al., 2016)

$$\max_F \mathbb{E}_\nu[\min(0, -1 + F(\boldsymbol{X}))] + \mathbb{E}_{\mu_z}[\min(0, -1 - F(G(\boldsymbol{Z})))] \tag{57}$$

$$\min_G \|\mathbb{E}_\nu[\phi(\boldsymbol{X})] - \mathbb{E}_{\mu_z}[\phi(G(\boldsymbol{Z}))]\|^2 \tag{58}$$

$F$ is a linear function of $\phi$ ($\exists w$ with $w^T \phi = F$)

## C.3 Network Architectures

Table 2: The architecture of DCGAN(Radford et al., 2016) for image Generation experiments on CIFAR-10 and CIFAR-100. The slopes of all leaky-ReLU functions in the networks were set to 0.2.

| $z \in \mathbb{R}^{128} \sim \mathcal{N}(0, I)$ |
| --- |
| dense $\to M_g \times M_g \times 512$ |
| 4×4, stride=2 deconv. BN 256 ReLU |
| 4×4, stride=2 deconv. BN 128 ReLU |
| 4×4, stride=2 deconv. BN 64 ReLU |
| 3×3, stride=1 conv. 3 Tanh |

(a) Generator, $M_g = 4$ for CIFAR-10 and CIFAR-100

| RGB image $\boldsymbol{x} \in \mathbb{R}^{M \times M \times 3}$ |
| --- |
| 3×3, stride=1 conv 64 lReLU
4×4, stride=2 conv 64 lReLU |
| 3×3, stride=1 conv 128 lReLU
4×4, stride=2 conv 128 lReLU |
| 3×3, stride=1 conv 256 lReLU
4×4, stride=2 conv 256 lReLU |
| 3×3, stride=1 conv. 512 lReLU |
| dense $\to 1$ |

(b) Discriminator, $M = 32$ for CIFAR10 and CIFAR-100

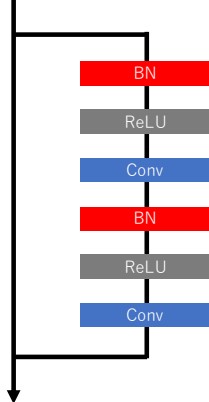

Fig 6: Resblock architectures for CIFAR-10 and CIFAR-100. We used similar architectures as the ones used in Gulrajani et al. (2017)

Table 3: ResNet architectures for CIFAR-10 and CIFAR-100. We used similar architectures as the ones used in Gulrajani et al. (2017).Resblock model is Fig 6

| $z \in \mathbb{R}^{128} \sim \mathcal{N}(0, I)$ |
| :---: |
| dense, $4 \times 4 \times 128$ |
| ResBlock up 128 |
| ResBlock up 128 |
| ResBlock up 128 |
| BN, ReLU, $3 \times 3$ conv, 3 Tanh |
| (a) Generator |

| RGB image $\boldsymbol{x} \in \mathbb{R}^{32 \times 32 \times 3}$ |
| :---: |
| ResBlock down 128 |
| ResBlock down 128 |
| ResBlock 128 |
| ResBlock 128 |
| ReLU |
| Global sum pooling |
| dense $\rightarrow 1$ |
| (b) Discriminator |

Table 4: ResNet generator architectures (large version) for CIFAR-10 and CIFAR-100.Resblock model is Fig 6

| $z \in \mathbb{R}^{128} \sim \mathcal{N}(0, I)$ |
| :---: |
| dense, $4 \times 4 \times 128$ |
| ResBlock up 256 |
| ResBlock up 256 |
| ResBlock up 256 |
| BN, ReLU, $3 \times 3$ conv, 3 Tanh |
| (a) Generator |

Table 5: ResNet architectures for image generation on ImageNet dataset.Resblock model is Fig 6

| $z \in \mathbb{R}^{128} \sim \mathcal{N}(0, I)$ |
| --- |
| dense, $4 \times 4 \times 1024$ |
| ResBlock up 1024 |
| ResBlock up 512 |
| ResBlock up 256 |
| ResBlock up 128 |
| ResBlock up 64 |
| BN, ReLU, 3×3 conv 3 |
| Tanh |

(a) Generator

| RGB image $x \in \mathbb{R}^{64 \times 64 \times 3}$ |
| --- |
| ResBlock down 64 |
| ResBlock down 128 |
| ResBlock down 256 |
| ResBlock down 512 |
| ResBlock down 1024 |
| ResBlock 1024 |
| ReLU |
| Global sum pooling |
| dense $\to 1$ |

(b) Discriminator for unconditional GANs.

Table 6: EGAN(Dai et al., 2017)'s decoder models used in our experiments on CIFAR-10 and CIFAR-100 . The slopes of all lReLU functions in the networks were set to 0.2. Fig 6 illustrates our Resblock model.

| RGB image $x \in \mathbb{R}^{M \times M \times 3}$ |
| --- |
| 3×3, stride=1 conv 64 lReLU
4×4, stride=2 conv 64 lReLU |
| 3×3, stride=1 conv 128 lReLU
4×4, stride=2 conv 128 lReLU |
| 3×3, stride=1 conv 256 lReLU
4×4, stride=2 conv 256 lReLU |
| 3×3, stride=1 conv. 512 lReLU |
| dense $\to$ 128 * 2 |

(a) Convolution decoder, $M = 32$ for CIFAR10 and CIFAR-100

| RGB image $x \in \mathbb{R}^{32 \times 32 \times 3}$ |
| --- |
| ResBlock down 64 |
| ResBlock down 128 |
| ResBlock down 256 |
| ResBlock down 512 |
| ResBlock down 1024 |
| ResBlock 1024 |
| ReLU |
| Global sum pooling |
| dense $\to$ 128 * 2 |

(b) ResNet decoder.

## C.4 DETAILS FOR THE EXPERIMENTS ON CIFAR-10 AND CIFAR-100

**Experimental setting for the evaluation of the method's robustness against the choice of objective functions**

For the model, we chose the architecture of DCGAN(Radford et al., 2016) (Table 2) and ResNet (Table 3) with spectral normalization applied to full connect layer and convolution layer(SNDCGAN, SNResNet). For the optimization, we used Adam(Kingma & Ba, 2015) and chose ($\alpha = 0.0002, \beta_1 = 0, \beta_2 = 0.9$) for the hyperparameters. Also, we chose $n_{dis} = 1, n_{gen} = 1$ for SNDCGAN and ($n_{dis} = 5, n_{gen} = 1$) for SNResNet. We updated the generator $100k$ times and linearly decayed the learning rate over last $5k$ iterations.

**Experimental setting for the evaluation of the method's robustness against the choice of the prior dimension**

We chose SNDCGAN for the model, and optimized the network with Adam using the hyperparameter ($\alpha = 0.0002, \beta_1 = 0, \beta_2 = 0.9$). For the update of the discriminator and the generator, we set $n_{dis} = 1, n_{gen} = 1$. We trained the generator $50k$ times and linearly decayed the learning rate over last $5k$ iterations. For the objective function, we chose *GAN-variant2* in Appendix C.2. We also set $\lambda = 3.0, d = 0.01$ for the parameters in (13). For this set of the experiments, we repeated the experiments three times with different seeds, and reported the maximum, mean and minimum. We tested with prior dimensions of range $\dim(\boldsymbol{z}) = 3, 4, 5, 7, 10, 500, 1000, 2000, 4000, 6000, 8000, 10000$.

**Comparison with EGAN and other methods**

For the model, we chose SNDCGAN , SNResNet and SNResNetLarge, and trained the networks using Adam with hyperparameters ($\alpha = 0.0002, \beta_1 = 0, \beta_2 = 0.9$). SNResNetLarge is a model that was used in (Miyato et al., 2018). It is a same model as SNResNet except that it is uses a larger generator (Table 4). For both models, We trained the generator $100k$ times and linearly decayed the learning rate over last $10k$ iterations. For EGAN-Ent-VI (Dai et al., 2017), we used an additional decoder equipped with convolution layers. We used $n_{dis} = 1, n_{gen} = 1, n_{dec} = 5$ for SNDCGAN, and $n_{dis} = 1, n_{gen} = 5, n_{dec} = 5$ for SNResNet. The table below is the list of the choices of the hyperparameters ($\lambda, d$) in equation (13). we used in our comparative study, sorted by models.

Table 7: List of the hyperparameter choices.

| Method | objective | $n_{dis}$ | $\lambda$ | $d$ |
|---|---|---|---|---|
| SNDCGAN + DC reg (CIFAR-10) | GAN-variant2 | 1 | 3 | 0 |
| SNDCGAN + DC reg (CIFAR-100) | GAN-variant2 | 1 | 3 | 0 |
| SNDCGAN-hinge + DC reg (CIFAR-10) | GAN-hinge | 1 | 3 | 0 |
| SNDCGAN-hinge + DC reg (CIFAR-100) | GAN-hinge | 1 | 3 | 0 |
| SNResNet + DC reg (CIFAR-10) | GAN-variant2 | 3 | 3 | 0 |
| SNResNet + DC reg (CIFAR-100) | GAN-variant2 | 5 | 3 | 0 |
| SNResNetLarge + DC reg (CIFAR-10) | GAN-variant2 | 5 | 3 | 0 |
| SNResNetLarge + DC reg (CIFAR-100) | GAN-variant2 | 5 | 4 | 0 |
| SNResNetLarge-hinge + DC reg (CIFAR-10) | GAN-hinge | 5 | 4 | 0.01 |
| SNResNetLarge-hinge + DC reg (CIFAR-100) | GAN-hinge | 5 | 4 | 0.01 |

## C.5 IMAGE GENERATION ON IMAGENET

The images used in this set of experiments were resized to $64 \times 64$ pixels. The details of the architecture are given in Table 5. For the optimization, we used Adam with the same hyperparameters we used for ResNet on CIFAR-10 and CIFAR-100 dataset. We trained the networks with 250K generator updates, and applied linear decay for the learning rate after 200K iterations so that the rate would be 0 at the end. We set $\lambda = 6.0, d = 0.01$ for the parameters in equation (13).

# D    APPENDIX RESULTS

## D.1    APPENDIX RESULT ON ARTIFICIAL DATA

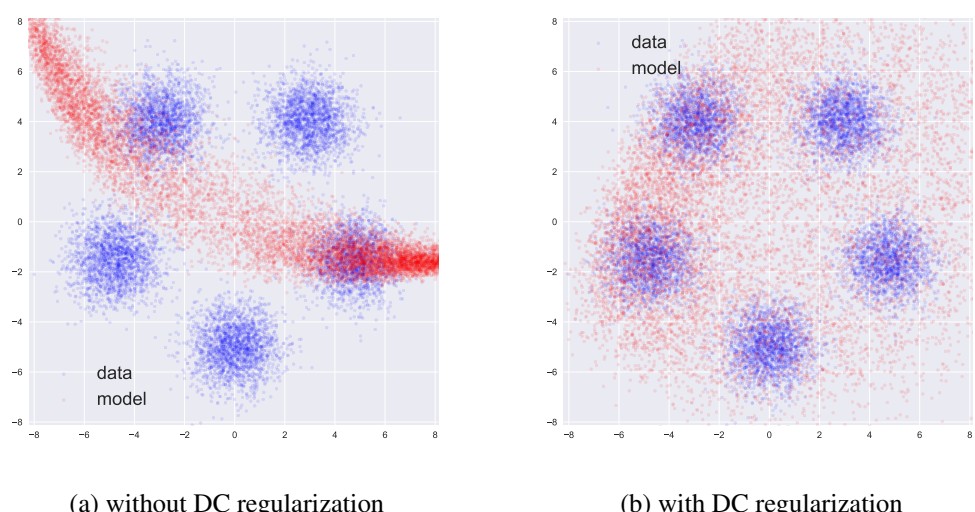

(a) without DC regularization                    (b) with DC regularization

Fig 7:  Generator samples on GMM fitting by GANs (a) without and (b) with DC regularization

## D.2    APPENDIX RESULT ON CIFAR-10 AND CIFAR-100

**Experiment with different architectures**

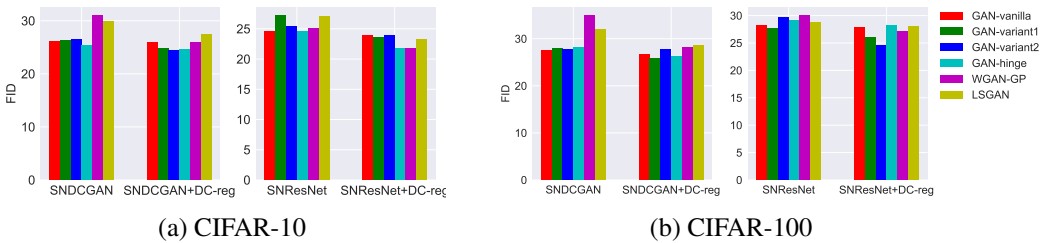

(a) CIFAR-10                              (b) CIFAR-100

Fig 8:  FIDs for unsupervised image generation on CIFAR-10 and CIFAR-100 (lower the better).

**Experiment with different architectures on WGAN-GP**

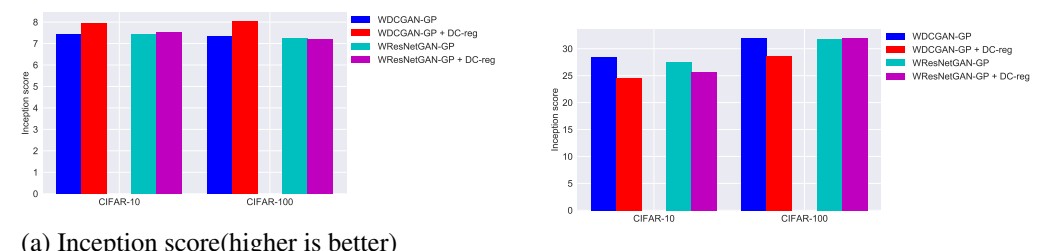

(b) FID(lower is better)

(a) Inception score(higher is better)

Fig 9: Results of WGAN-GP

**Experiment with varying prior dimension**

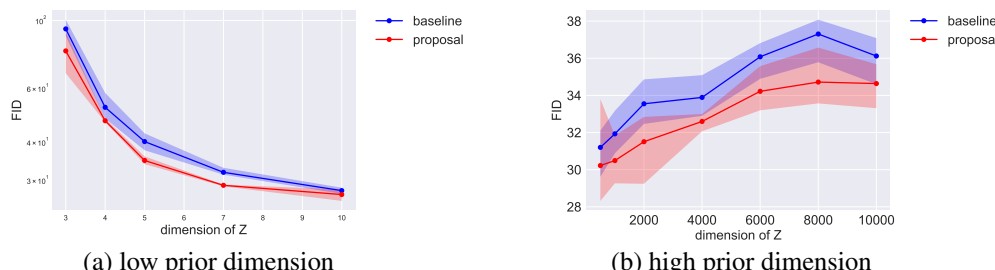

(a) low prior dimension                    (b) high prior dimension

Fig 10:  FID performance of DC regularization on low dimensional prior and high dimensional prior.

**Comparison with other methods**

Table 8:  Inception scores and FIDs with unsupervised image generation on CIFAR-10 and CIFAR-100. CIFAR-10 results for the models designated with † are cited from (Miyato et al., 2018), and the CIFAR-10 results results with ‡ are cited from (Karras et al., 2018).  For the details of the objective functions, see Appendix C.2.

| Method | Inception score | | FID | |
|---|---|---|---|---|
| | CIFAR-10 | CIFAR-100 | CIFAR-10 | CIFAR-100 |
| Real data | 11.24 | 14.79 | 7.6 | 8.94 |
| EGAN-Ent-VI(SNDCGAN) | 6.95±.08 | 6.62±.10 | 29.0 | 33.3 |
| EGAN-Ent-VI(SNResNet) | 7.31±.12 | 6.67±.10 | 27.0 | 30.5 |
| feature matching(SNDCGAN) | 7.54±.10 | 7.71±.06 | 25.9 | 29.2 |
| **proposal** | | | | |
| SNDCGAN + DC reg | 8.08±.12 | 8.12±.11 | 24.6 | 25.8 |
| SNDCGAN-hinge + DC reg | 7.70±.11 | 7.99±.09 | 24.7 | 26.1 |
| SNResNet + DC reg | 8.27±.08 | 8.27±.13 | 24.3 | 24.6 |
| SNResnetLarge + DC reg | 8.41±.10 | 8.20±.08 | 20.6 | 24.8 |
| SNResnetLarge-hinge + DC reg | 8.29±.09 | **8.41**±.11 | **19.5** | **23.6** |
| **baseline** | | | | |
| SNDCGAN† | 7.42±.08 | 7.74±.08 | 29.3 | 27.9 |
| SNDCGAN-hinge† | 7.58±.12 | 7.57±.07 | 25.5 | 28.1 |
| SNResnetLarge-hinge† | 8.22±.05 | 7.54±.13 | 21.7 | 26.6 |
| Progressive GANs‡ | **8.56**±.06 | | | |

**Image generation on CIFAR-10 and CIFAR-100**

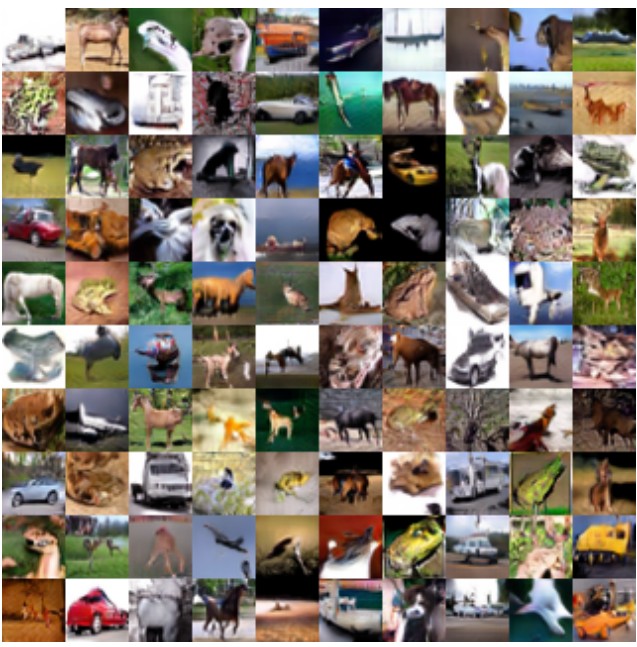

(a) image generation of CIFAR-10

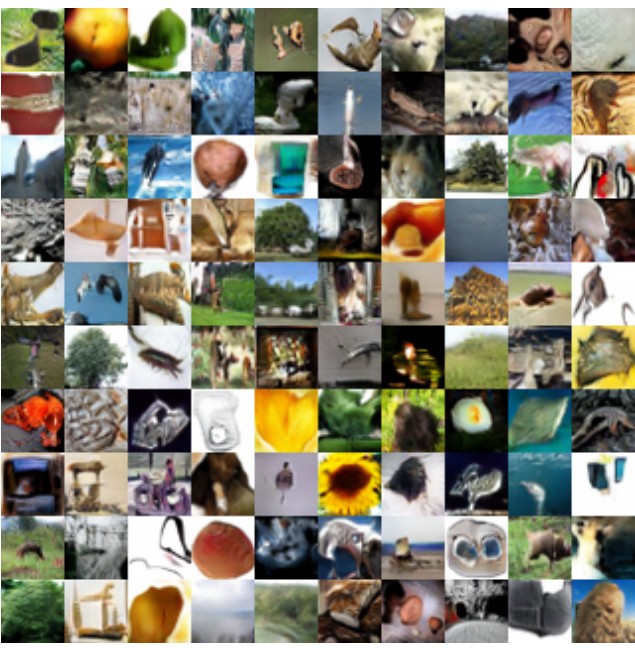

(b) image generation of CIFAR-100

Fig 11: Image generation of CIFAR-10 and CIFAR-100

### D.3 APPENDIX RESULT ON IMAGENET

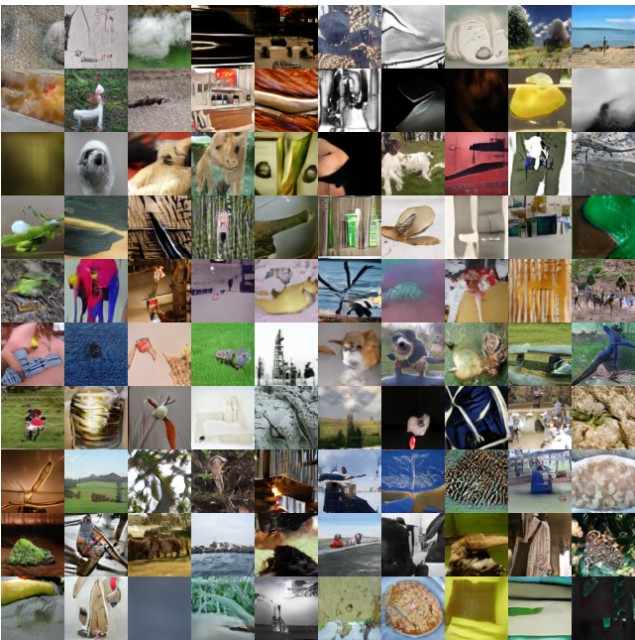

(a) image generation of baseline model

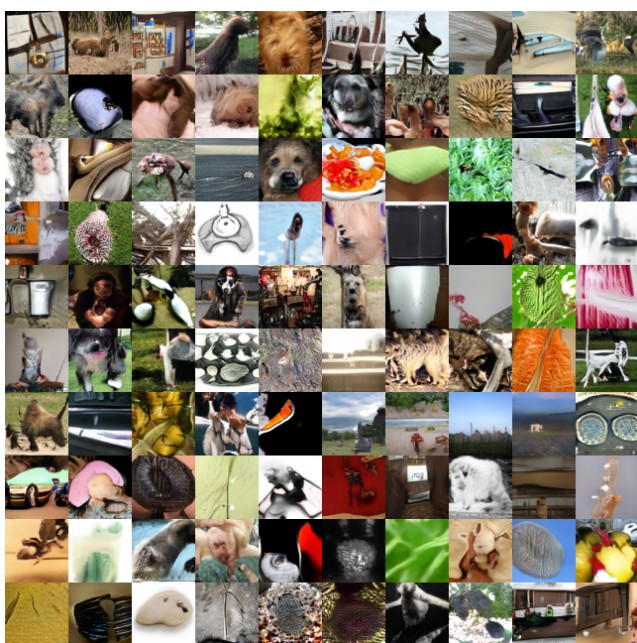

(b) image generation of proposal model

Fig 12: Image generation of ImageNet

