# OpenReview forum: "DISTRIBUTIONAL CONCAVITY REGULARIZATION FOR GANS"
_ICLR.cc/2019/Conference_

### Official Review · AnonReviewer1 · 2018-11-02
**Nice experimental paper (with theory backing)**

**Rating:** 7
**Confidence:** 1

**Review:**

In this paper, the authors claim that they are able to update the generator better to avoid generator mode collapse and also increase the stability of GANs training by indirectly increasing the entropy of the generator until it matches the entropy of the original data distribution using functional gradient methods.

The paper is interesting and well written. However, there is a lot of work coming out in the field of GANs currently, so I am not able to comment on the novelty of this regularization approach, and I am interested to know how this method performs when compared to other techniques to avoid mode collapse such as feature matching and mini-batch discrimination, etc.

---

> ### Author Response · Authors · 2018-11-09
> **Thank you very much for the suggestions and comment.**
>
> Thank you very much for the comment.  We believe that the core novelty of our work is in introducing a type of regularization based on a functional gradient.  Because we were able to conduct additional experiment in time, we also conducted feature matching as well and confirmed the superiority of DC-regularization over the method (Table 8).

---

### Official Review · AnonReviewer3 · 2018-11-02
**potentially useful heuristic for GANs with vague maths**

**Rating:** 6
**Confidence:** 4

**Review:**

GANs (generative adversarial network) represent a recently introduced min-max generative modelling scheme with several successful applications. Unfortunately, GANs often show unstable behaviour during the training phase. The authors of the submission propose a functional-gradient type entropy-promoting approach to tackle this problem, as estimating entropy is computationally difficult.

While the idea of the submission might be useful in some applications, the work is rather vaguely written, it is in draft phase:
1. Abbreviations, notations are not defined: GAN, WGAN-GP, DNN, FID (the complete name only shows up in Section 4), softplus, sigmoid, D_{\theta_{old}}, ...
2. While the primary motivation of the work is claimed to be 'mode collapse', it does not turn out from the submission what mode collapse is.
3. Estimating entropies is a standard practice in statistics and machine learning, with an arsenal of estimators; the motivation of the submission is questionable.
4. Differentiation w.r.t. functions (or more generally elements in normed spaces) is a well-defined concept in mathematics, including the notions of Gateaux, Frechet and Hadamard differentiability. It is not clear why the authors neglect these classical concepts, and are talking about 'random functional perturbations', ... It is also unclear where the optimized transformation (T) lives; the authors are trying to differentiate over some function space which is undefined.

While the idea of the work might be useful in practice, the current submission requires significant revision and work before publication.

---

After paper revisions:

Thank you for the updates. The submission definitely improved. I have changed my score to '6: Marginally above acceptance threshold'; the suggested regularization can be a useful heuristic for the GAN community.

---

> ### Author Response · Authors · 2018-11-09
> **Thank you very much for the suggestions and comment for the revisions!**
>
> Thank you very much for the comment.  We have revised the script to reflect the suggestions, and we would like to articulate on the changes we have made.   Because we are out of space, we will provide the  answers over several comments.  For the list of mentioned references, please see the 'last' comment.
>
> > 1. Abbreviations, notations are not defined: GAN, WGAN-GP, DNN, FID (the complete name only shows up in Section 4), softplus, sigmoid, D_{\theta_{old}}, …
>
> We clarified the abbreviations for commonly used acronyms, such as Deep Neural Networks and Frechet inception distance, and gave definitions to undefined  symbols.
>
> > 2. While the primary motivation of the work is claimed to be 'mode collapse', it does not turn out from the submission what mode collapse is.
>
> We have given more detailed explanation of mode collapse, and provided citation that can be consulted for further information regarding its definition. In short, Mode collapse collectively refers to the lack of diversity in generator distribution [1-5]. This phenomena happens even in the simplest case of generating a mixture of Gaussian Distribution (See Fig 22, [1] for example) . Without any notable countermeasure, GANs tend to produce a distribution with less number of modes than the target distribution .
>
> > 3. Estimating entropies is a standard practice in statistics and machine learning, with an arsenal of estimators; the motivation of the submission is questionable.
>
> In the revision, we have emphasized how the classical empirical techniques in estimating the entropy are not suited for the training of GANs aimed toward the synthesis of high dimensional distribution. In fact, our method yields much better performance than Energy-based GANs(EGANs) [6], that uses a rather high calibre variational inference based technique to directly estimate the entropy, which was, in their experiment, performed the best among all classical techniques they tested. For more details, please consult the original text.
>
> In many application of GANs, the objective is to synthesize a generative model on high dimensional space, and it is a very difficult problem on its own to rely on classical techniques to estimate the entropy in such high dimensional space. In most study of generative models, the distribution is defined in term of latent variables, and its law is given by G# p where p is some known distribution---or the pushforward of p with a measurable function G with an euclidean range.  The most straightforward approach in entropy estimation uses some form of plugin Estimator based on Kernel density estimation and nearest neighbor estimation hat p:
> estimate  =  g(\hat p(X_i), X_i  \in Data ).
> For the real world applications of GANs (e.g image generations), the dimensions of the data space are high.  On the CIFAR 10 benchmark, the dimension is   32*32*3, and the dimension of the ImageNet benchmark is high as 64*64*3.   Even for Kernel density estimator on of  regular density on d dimensional Euclidean space for example, the mean square error is MSE(f*, \hat f) = O(h^4  +  1/(n h^{d})) where h is the bandwidth, n is the number of samples [7,8].
> This is to say we need small enough bandwidth with tremendous size of n for large d. On the top of this, for  the density of p_g of G\# p is computed through the formula
> H(p_g) = H(p(z)) + E_p(z)[det(D_z G(z))].
> In order to compute E_p(z)[det(D_z G(z))], we need at least O(n d^2) (often d^3 in usual implementation)  and this expression does not include the massicve scaler that is dependent on the number of parameters, which again, is  often extremely large, and can be in order of millions（Fig 2 in [9]) .  Again, Energy-based GANs(EGANs)[6], that uses a rather high calibre variational inference based technique are not performing too well relative to our approach.
> Also, one of the fundamental motivation for the development of GANs techniques is to do away with the precise and explicit density of the target distribution in high dimensional space. For more details of these motives, please consult classical literatures in GANs [1,4,10].

---

> > ### Author Response · Authors · 2018-11-09
> > **response, con'd**
> >
> > > 4. Differentiation w.r.t. functions (or more generally elements in normed spaces) is a well-defined concept in mathematics, including the notions of Gateaux, Frechet and Hadamard differentiability. It is not clear why the authors neglect these classical concepts, and are talking about 'random functional perturbations', ... It is also unclear where the optimized transformation (T) lives; the authors are trying to differentiate over some function space which is undefined.
> >
> > We have added more descriptions in the theory section and expressed our intention that we are taking Frechet derivatives of a functional defined on a Hilbert space consisting of functions that are L2 integrable with respect to the probability measure of concern. In functional gradient applications to generative models [11, 12],  conditions required for  the Frechet differentiability of the objective functional are often assumed to hold. We hope that our revisions made the paper more readable to the wider audience, and hope that  the paper now assumes less knowledge of GANs in understanding our idea.
> >
> > [1] Ian Goodfellow. NIPS 2016 tutorial: Generative adversarial networks. arXiv preprint arXiv:1701.00160, 2016.
> > [2] Luke Metz, Ben Poole, David Pfau, and Jascha Sohl-Dickstein. Unrolled generative adversarial networks. In ICLR, 2017.
> > [3] Martin Arjovsky and Le ́on Bottou. Towards principled methods for training generative adversarial networks. In ICLR, 2017.
> > [4] Martin Arjovsky, Soumith Chintala, and Le ́on Bottou. Wasserstein generative adversarial networks. In ICML, pp. 214–223, 2017.
> > [5] Zinan Lin, Ashish Khetan, Giulia Fanti, and Sewoong Oh. Pac gan: The power of two samples in generative adversarial networks. arXiv preprint arXiv:1712.04086, 2017.
> > [6] Zihang Dai, Amjad Almahairi, Philip Bachman, Eduard Hovy, and Aaron Courville. Calibrating energy-based generative adversarial networks. In ICLR, 2017.
> > [7] Theophilos Cacoullos. Estimation of a multivariate density. Annals of the Institute of Statistical Mathematics, 18(1):179–189, 1966.
> > [8] Arkadas Ozakin and Alexander G Gray. Submanifold density estimation. In Advances in Neural Information Processing Systems, pp. 1375–1382, 2009.
> > [9] Alfredo Canziani, Eugenio Culurciello, Adam Paszke. AN ANALYSIS OF DEEP NEURAL NETWORK MODELS FOR PRACTICAL APPLICATIONS. arXiv preprint arXiv:1605.07678, 2017.
> > [10] Ian Goodfellow, Jean Pouget-Abadie, Mehdi Mirza, Bing Xu, David Warde-Farley, Sherjil Ozair, Aaron Courville, and Yoshua Bengio. Generative adversarial nets. In NIPS, pp. 2672–2680, 2014.
> > [11] Atsushi Nitanda and Taiji Suzuki. Gradient layer: Enhancing the convergence of adversarial training for generative models. In AISTATS, pp. 1008–1016, 2018.
> > [12] Rie Johnson and Tong Zhang. Composite functional gradient learning of generative adversarial models. In ICML, pp. 2376–2384, 2018.

---

### Official Review · AnonReviewer2 · 2018-11-04
**A good paper**

**Rating:** 8
**Confidence:** 1

**Review:**

The authors make use of the theory of functional gradient, based on optimal transport, to develop a method that can promote the entropy of the generator distribution without directly estimating the entropy itself. Theoretical results are provided as well as necessary experiments to support their technique's outperformance in some data sets. I found that this is an interesting paper, both original ideal and numerical results.

---

> ### Author Response · Authors · 2018-11-09
> **Thank you!**
>
> Thank you very much for the comment.  We believe that functional gradient based methods have much room to explore, and it is our hope that many aspects of GANs can be analyzed using this philosophy.

---

### Official Review · AnonReviewer4 · 2018-11-12
**Sound method and good results**

**Rating:** 7
**Confidence:** 4

**Review:**

Summary:
This paper proposes distributional concavity regularization for GANs which encourages producing generator distributions with higher entropy. The paper motivates the proposed method as follows:
-       Using the concept of functional gradient, the paper interprets the update in the generator parameters as an update in the generator distribution
-       Given this functional gradient perspective, the paper proposes updating the generator distribution toward a target distribution which has *higher entropy and satisfies monoticity*
-       Then, the paper proves that this condition can be satisfied by ensuring that generator’s objective (L) is concave
-       Since it’s difficult to ensure concavity when parametrizing generators as deep neural networks, the paper proposes adding a simple penalty term that encourages the concavity of generator objective
Experiments confirm the validity the proposed approach. Interestingly, the paper shows that performance of multiple GAN variants can be improved with their proposed method on several image datasets

Strengths:
-   	The proposed method is very interesting and is based on sound theory
-   	Connection to optimal transport theory is also interesting
-   	In practice, the method is very simple to implement and seems to produce good results

Weaknesses:
-       Readability of the paper can be generally improved. I had to go over the paper many times to get the idea.
-       Figures should be provided with more detailed captions, which explain main result and providing context (e.g. explaining baselines).

Questions/Comments:
-       Equation (7) has typos (uses theta_old instead of theta in some places)
-       Section 4.1 (effect of monoticity) is a bit confusing. My understanding is that parameter update rule of equation (3) and (6) are equivalent, but you seem to use (6) there. Can you clarify what you do there and in general this experiment a bit more?
-       Comparing with entropy maximization method of EGAN (Dai et al, 2017) is a good idea, but I’m wondering if you can compare it on low dimensional settings (e.g. as in Fig 2). It is also not clear why increasing entropy with EGAN-VI is worse than baselines in Table 1.


Overall recommendation:
The paper is based on sound theory and provides very interesting perspective. The method seems to work in practice on a variety of experimental setting. Therefore, I recommend accepting it.

---

> ### Comment · AnonReviewer4 · 2018-11-13
> **Emphasize improving readability of the paper**
>
> I would like to emphasize that the main weakness in the paper, in my opinion, is that it can be quite hard to read for the general ICLR community. The authors are strongly encouraged to try to make it more accessible, which will in fact will increase the impact of the paper eventually.

---

> > ### Author Response · Authors · 2018-11-21
> > **Thank you very much for the positive and helpful comments!**
> >
> >
> >  As advised,  we made particular effort to improve the readability of the paper.  We made multiple large/small revisions throughout the paper to reflect the suggestions.  We would like to elaborate on our revisions  in the form of writing a response to each comment we received:
> >
> > >Weaknesses:
> > >-Readability of the paper can be generally improved. I had to go over the paper many times to get the idea.
> >
> > We  took this advice as seriously as possible, and particularly  reorganized introduction and section 2.
> > We reworded many phrases in an effort to convey our ideas from more intuitive perspective.
> > We moved the technical description of the section 2 to the appendix for the readers with interests in functional theoretic background of our algorithm.We hope that our revision improves the readability of the paper.
> >
> > >-Figures should be provided with more detailed captions, which explain main result and providing context
> > >(e.g. explaining baselines).
> >
> > We added more description to the captions, and elaborated on the experiment described by each figure.
> >
> > >Questions/Comments:
> > >-Equation (7) has typos (uses theta_old instead of theta in some places)
> >
> > Thank you  very much!  We fixed the typos in the equation 7.
> >
> > >-Section 4.1 (effect of monoticity) is a bit confusing. My understanding is that parameter update rule of
> > >equation (3) and (6) are equivalent, but you seem to use (6) there. Can you clarify what you do there and in
> > >general this experiment a bit more?
> >
> > We must admit that we were not clear enough about the motive of the second experiment in section 4.1.
> > As we have additionally explained in the revised version of the section 2.3,  the “monotoniciy“ is a property satisfied by the optimal-transport-based update that can possibly have a good  effect on the distillation step.
> > Monotonicity is a property that our algorithm guarantees for the map used in our update as well.
> > In distillation step, the goal of the user is to  approximate the target distribution with the parametric distribution,
> > and as many SGD steps can be used as liked. In conventional GANs, only one SGD update is applied to the parametric generator G.
> > The purpose of our second experiment in section 4.1 is to assess the effect of monotonicity on this distillation step.  We prepared a pair of target distributions---one constructed with a monotonic map and another constructed with a non-monotonic map, with the former being further away(in Wasserstein sense) from the current distribution and both of them yielding the same value for the objective function.
> > Against the intuition based on “the distance”,  the distillation procedure is easier for the distribution constructed with the monotonic map.
> > This experiments demonstrates a case in which the monotonicity works in favor of the training of G( in distillation step).
> >
> > >-Comparing with entropy maximization method of EGAN (Dai et al, 2017) is a good idea, but I’m wondering
> > >if you can compare it on low dimensional settings (e.g. as in Fig 2). It is also not clear why increasing entropy
> > >with EGAN-VI is worse than baselines in Table 1.
> >
> > We conducted experiments on the low dimensional setting as well, and confirmed that our implementation of EGAN-VI indeed achieves higher entropy than the vanilla experiment without the regularization. As we can see in the results, the  performance is not  too impressive, however.
> >
> > As for the second concern that is being raised, we would like to note that the original implementation of EGAN in the publication was conducted without Spectral Normalization(SN), which generally improves the results for most methods. Our baseline method is not an ordinary DCGAN, but the DCGAN with SN that is known to perform at competitive quality on the dataset like ImageNet and CIFAR10.
> > In fact, the “vanilla DCGAN with SN” and “EGAN with SN” both perform better than the vanilla EGAN as well. In this light, it is not so surprising that  EGAN with SN  performs worse than the vanilla SN-DCGAN on CIFAR10,  because the variational inference for the entropy of the distribution in high dimensional space like the one dealt with in CIFAR10.
> >  For the experiment, we used EGAN-VI based on Gaussian distribution, as opposed to EGAN-Nearest Neighbor. In the original publication, this version of  EGAN-VI was being used for their experiment on CIFAR10.   We experimented with multiple parameters and always reported the result with best Inception Score/ FID.
> > In general, EGAN needs to prepare a decoder in addition to the pair of Generator and Discriminator. Because the training for both of them are being conducted separately, it is difficult to find the right balance between the two during the training.

---

### Meta-Review · Area_Chair1 · 2018-12-18
**A solid contribution to regularize GANs**

**Confidence:** 4
**Recommendation:** Accept (Poster)

**Metareview:**

This paper proposes distributional concavity regularization for GANs which encourages producing generator distributions with higher entropy.

The reviewers found the contribution interesting for the ICLR community. R3 initially found the paper lacked clarity, but the authors took the feedback in consideration and made significant improvements in their revision. The reviewers all agreed that the updated paper should be accepted.